# Formulation of Amphotericin B in PEGylated Liposomes for Improved Treatment of Cutaneous Leishmaniasis by Parenteral and Oral Routes

**DOI:** 10.3390/pharmaceutics14050989

**Published:** 2022-05-05

**Authors:** Guilherme S. Ramos, Virgínia M. R. Vallejos, Gabriel S. M. Borges, Raquel M. Almeida, Izabela M. Alves, Marta M. G. Aguiar, Christian Fernandes, Pedro P. G. Guimarães, Ricardo T. Fujiwara, Philippe M. Loiseau, Lucas A. M. Ferreira, Frédéric Frézard

**Affiliations:** 1Department of Physiology and Biophysics, Institute of Biological Sciences, Universidade Federal de Minas Gerais, Belo Horizonte 31270-901, MG, Brazil; gramos@ufmg.br (G.S.R.); virginia.carregal@uemg.br (V.M.R.V.); ppiresgo@reitoria.ufmg.br (P.P.G.G.); 2Faculty of Pharmacy, Universidade Federal de Minas Gerais, Belo Horizonte 31270-901, MG, Brazil; gsmb@ufmg.br (G.S.M.B.); izabela.alves@ebserh.gov.br (I.M.A.); martagontijo@ufmg.br (M.M.G.A.); cfernandes@farmacia.ufmg.br (C.F.); lucas@farmacia.ufmg.br (L.A.M.F.); 3Department of Parasitology, Institute of Biological Sciences, Universidade Federal de Minas Gerais, Belo Horizonte 31270-901, MG, Brazil; raquel-martins@ufmg.br (R.M.A.); fujiwara@icb.ufmg.br (R.T.F.); 4Faculty of Pharmacy, Antiparasite Chemotherapy, UMR 8076 CNRS BioCIS, University Paris-Saclay, F-92296 Chatenay-Malabry, France; philippe.loiseau@universite-paris-saclay.fr

**Keywords:** liposomes, amphotericin B, leishmaniasis, oral route, PEGylation, cutaneous leishmaniasis

## Abstract

Liposomal amphotericin B (AmB) or AmBisome^®^ is the most effective and safe therapeutic agent for visceral leishmaniasis (VL), but its clinical efficacy is limited in cutaneous leishmaniasis (CL) and HIV/VL co-infection. The aim of this work was to develop a formulation of AmB in PEGylated liposomes and compare its efficacy to AmBisome^®^ in a murine model of CL. Formulations of AmB in conventional and PEGylated liposomes were characterized for particle size and morphology, drug encapsulation efficiency and aggregation state. Those were compared to AmBisome^®^ in *Leishmania amazonensis*-infected BALB/c mice for their effects on the lesion size growth and parasite load. The conventional and PEGylated formulations showed vesicles with 100–130 nm diameter and low polydispersity, incorporating more than 95% of AmB under the non-aggregated form. Following parenteral administration in the murine model of CL, the PEGylated formulation of AmB significantly reduced the lesion size growth and parasite load, in comparison to control groups, in contrast to conventional liposomal AmB. The PEGylated formulation of AmB was also effective when given by oral route on a 2-day regimen. This work reports for the first time that PEGylated liposomal AmB can improve the treatment of experimental cutaneous leishmaniasis by both parenteral and oral routes.

## 1. Introduction

Cutaneous and muco-cutaneous leishmaniases (CL/MCL) are disfiguring diseases caused by protozoan parasites belonging to the genus *Leishmania*. An estimated 1 million new cases occur annually worldwide. CL and MCL are associated with population displacement, poor housing, lack of financial resources, malnutrition and a weak immune system. These features and the low interest of the pharmaceutical industry in developing new drugs for this group of diseases led to its classification as a neglected tropical disease [1]. Depending on the parasite species, the endemic region and the patient immunological status, clinical manifestations vary from a single nodular lesion to disseminated forms including MCL [1]. In the most frequent cases of simple CL with one or few lesions, local treatment with intralesional antimony or topical formulations of paromomycin is generally adequate. On the other hand, systemic treatment is recommended for complex CL with a high risk of mucosal involvement, numerous or very large lesions, disseminated forms, lesions location not compatible with local treatment, or immunosuppression [2].

Regarding the systemic treatment of tegumentary leishmaniases, pentavalent antimonials are still considered the “gold standard”, despite severe toxicities resulting in low patient compliance. Alternative systemic treatment options for complex CL are intravenous liposomal amphotericin B (AmB) or AmBisome^®^, intravenous or intramuscular pentamidine, oral miltefosine and oral azoles [3].

Though AmBisome^®^ shows at least a 95% cure rate for visceral leishmaniasis (VL) in the Indian subcontinent and in Southern Europe [4], clinical data for New World CL are scarce. Moderate efficacies with a cure rate lower than 80% have been reported [5,6,7]. The use of AmBisome^®^ is also limited by systemic side effects, most frequently acute infusion-related events and nephrotoxicity [3]. Other drawbacks that limit its use in developing countries are its high cost and the need for parenteral administration.

Recently, AmBisome^®^ was compared to micellar AmB (Fungizone^®^) and another liposomal AmB product marketed in India (Fungisome^®^), for antileishmanial efficacy and intralesional drug accumulation, after parenteral administration in a murine model of CL [8,9]. AmBisome^®^ promoted significantly greater antileishmanial efficacy and drug accumulation within the infected lesion. It was proposed that the higher stability of AmB incorporation and the smaller vesicle size in AmBisome^®^ may contribute to the longer drug permanence in the bloodstream and, likely, to enhanced extravasation through the leaky capillaries in the inflamed lesion skin. This data also suggests that long-circulating liposomes may achieve greater efficacy against CL or disseminated *Leishmania* infections, through parenteral administration.

In the last decade, much effort has been devoted to the search for novel drug delivery strategies for AmB, including ways to improve its bioavailability by the oral route [10,11]. The potential benefits of an oral formulation of AmB would be: (i) reduction in drug- and formulation-related side effects; (ii) out-patient treatment; (iii) treatment cost reduction; (iv) and improved patient compliance. As recently reviewed [10], despite unfavorable drug physicochemical characteristics, such as large molecular weight, amphoteric nature, very poor water and lipid solubilities, as well as acid lability, progress has been achieved towards the development of orally-effective AmB formulations. Several formulations of AmB showed promising results by the oral route in experimental models of VL: lipid-based formulations [12,13]; amphotericin B-carbon nanotube conjugate [14], chitosan-based nanoformulation [15,16]. However, there has been no report so far on the efficacy of these formulations in experimental models of CL.

In the present work, a new process is presented for the incorporation of AmB into pre-formed liposomes, which was successfully applied to conventional and PEGylated liposomes. We also tested the hypothesis that long-circulating PEGylated liposomes may be more effective than conventional liposomes or AmBisome^®^ against CL, using a murine model. The PEGylated liposomal formulation was further evaluated for its efficacy by the oral route. Our data shows that PEGylated liposomal AmB is more effective than AmBisome^®^ by both parenteral and oral routes, in murine CL.

## 2. Materials and Methods

### 2.1. Materials

Miltefosine and cholesterol (CHOL) were purchased from Sigma-Aldrich (St. Louis, MO, USA). Hydrogenated soybean phosphatidylcholine (HSPC), distearoylphosphatidylglycerol (DSPG) and distearoylphosphatidylethanolamine-PEG(2000) (DSPE-PEG) were obtained from Lipoid (Ludwigshafen, Germany). AmBisome^®^ was from Gilead Science Inc. (Foster City, CA, USA). AmB was gently donated by Cristália (Cotia, SP, Brazil).

### 2.2. Preparation of Liposomal AmB Formulations

AmB was incorporated into pre-formed conventional and PEGylated liposomes made from HSPC:CHOL:DSPG (5:2.5:2 molar ratio) and HSPC:CHOL:DSPG:DSPE-PEG 2000 (5:2.5:2:0.5 molar ratio), respectively.

Briefly, multilamellar liposomes were first prepared in deionized water at a final lipid concentration of 50 mM. These multilamellar liposomes were transformed into unilamellar vesicles through five freeze-thaw cycles, followed by repeated extrusions (10 times) across 100-nm pore size polycarbonate membranes (Lipex^®^ Extruder, Burnaby, BC, Canada). For drug incorporation, AmB was first solubilized at 12.5 mg/mL with NaOH 0.1 M and protected from light. Immediately after solubilization, the solution was added to the pre-warmed (60 °C) liposome suspension at a 1:10 AmB/lipid molar ratio. After 2-min incubation at 60 °C under magnetic stirring, the pH of the mixture was decreased to 6.5 through the addition of an aliquot of 0.1 M acid HEPES solution. The drug/liposome mixture was further incubated for 5 min at 60 °C. A 180 g/L sucrose solution was then added at a sugar/lipid mass ratio of 2.8 and the mixture was immediately frozen in liquid nitrogen and freeze-dried for 48 h under the protection of light (freeze-dryer L101; Liotop, São Carlos, Brazil). The lyophilized samples were stored at 4 °C until use. Just before use, the samples were reconstituted by the addition of deionized water at a final AmB concentration of 4 mg/mL. Empty liposomes were prepared using the same protocol, except for the step of AmB incorporation. The formulations were further diluted in 5% dextrose for efficacy studies in animal models.

### 2.3. Characterization of Nanoparticle Size, Zeta Potential and Morphology

The mean hydrodynamic diameter, polydispersity index (PI) and zeta potential of the resulting liposome formulations were determined at 25 °C by dynamic light scattering (DLS) using a particle size analyzer (Zetasizer S90, Malvern, UK). The suspension was diluted 100 times in either 5% dextrose solution for particle size measurements or PBS (0.15 M NaCl, phosphate 10 mM, pH 7.2) for zeta potential measurements.

Liposome morphology was evaluated via cryogenic-transmission electron microscopy (Cryo-TEM). TEM samples were prepared on a carbon film-coated grid to which 1% sodium phosphotungstate was added for negative staining before images were obtained. Cryo-TEM specimens were prepared in the controlled environment vitrification system at 25 °C and ~100% relative humidity. Vitrified samples were examined with a Tecnai G2-20—FEI SuperTwin 200 kV at the Center of Microscopy at UFMG.

### 2.4. Determination of AmB Encapsulation Efficiency and AmB Total Content

AmB concentration in liposomal formulations (before and after filtration) was determined by a reverse phase HPLC method [17] with small modifications. Briefly, a Waters liquid chromatograph coupled with a 515 model Waters pump, a 717 plus automatic injector, and a TM 486 Waters DAD detector were used. A C_18_ (250 × 4.6 mm; 5 µm) column was used with a mixture of methanol, acetonitrile, tetrahydrofuran and 2.5 mM sodium edetate (42:18:10:30) as a mobile phase at a flow rate of 1 mL/min. The injection volume was 20 µL and detection was at 405 nm. The retention time for AmB was about 7 min.

The encapsulation efficiency (EE) of AmB in liposomes was evaluated based on the determination of AmB concentration before and after filtration with polyvinylidene fluoride membrane, 0.45 µm pore size (Millipore^®^, Burlington, MA, USA) [18]. The quantification of AmB before filtration provided the AmB content of the liposomes. The AmB that remained insoluble, and hence not properly incorporated into the nanocarrier, was removed by filtration. The EE was calculated as follows: 100 × [AmB] after filtration/[AmB] before filtration.

AmB total content (%) of the liposomes was obtained by comparing the area under the curve (AUC) of the AmB absorption peaks found before filtration of the liposomes and the AUC of AmB absorption peaks obtained from standard solutions of AmB. AmB total content (%) was calculated as follows: 100 × [AmB] before filtration / [AmB] standard solution. The final loading content of AmB was also calculated as the mass ratio of encapsulated AmB/lipids, in the formulation.

### 2.5. Evaluation of AmB Aggregation State

The different liposomal formulations were compared regarding the aggregation state of AmB by UV–visible spectroscopy and circular dichroism (CD) as described previously [19]. After reconstitution of liposomal AmB formulations with water, those were further diluted 1000-fold in PBS. The absorption and CD spectra were recorded in a 1.0-cm path length quartz cuvette in the range of 300–450 nm, just after dilution at 25 °C, under a nitrogen atmosphere on a computer-assisted Chirascan™ spectropolarimeter (Applied Photophysics, Leatherhead, UK). Data were obtained using an accumulation of three scans. The final spectra were obtained after subtraction of the PBS spectrum from the spectra of the samples. No signal was observed in the range of 300–450 nm for empty liposomal formulations. A solution of AmB was also prepared in PBS without liposomes, as a control, using the same process of preparation of the liposomal formulation, except for the freeze-drying step. All the data were expressed in terms of mdeg.

### 2.6. Stability of PEGylated Liposomal Formulation in Simulated Gastric Fluid

A simulated gastric fluid without enzyme (SGF) was prepared by dissolving 2 g de NaCl in water, adding 7 mL of HCl (37%) and completing the volume with water to 1 L. The pH was adjusted to pH 1.2 ± 0.1. The liposomal AmB formulations were diluted 10-fold in either the SGF or PBS and incubated at 37 °C for 2 h under constant stirring with a magnetic flea. CD and absorption spectra were registered as described in Section 2.5., after 100-fold dilution in PBS immediately after the incubation period. Particle size was also determined in the sample 100-fold diluted in PBS, using DLS as described in Section 2.3.

### 2.7. Antileishmanial Activity in Murine Model of CL

#### 2.7.1. Animals and Parasites

BALB/c mice (female, 4–6 weeks old, 18–22 g) were obtained from the central bioterium of the Federal University of Minas Gerais (UFMG, Belo Horizonte, Brazil). Free access to a standard diet was allowed, and tap water was supplied ad libitum.

The study involving animals was approved by the Ethical Committee for Animal Experimentation of the UFMG on 03/16/2020 with protocol number 54/2020.

The *Leishmania* strain used in the CL model was *Leishmania* (*L.*) *amazonensis* (IFLA/BR/1967/PH8), obtained from the cryopreservation bank of the Leishmania Biology Laboratory at ICB, UFMG. The cells were maintained in vitro as promastigotes at 24 ± 1 °C, pH 7.0, in Schneider’s Insect Medium (Gibco^®^; Thermo Fisher Scientific, Waltham, MA, USA) supplemented with 10% heat-inactivated bovine fetal serum (Cultilab, Brazil), 100 U/mL penicillin and 100 μg/mL streptomycin (Gibco^®^) in incubator Biochemical Oxygen Demand (BOD-Water-Jacketed Incubator, Thermo Scientific, Waltham, MA, USA). Promastigotes were grown in cell culture flasks of 25-mL volume (Corning Incorporated, Corning, NY, USA) with an initial inoculum of 1 × 10^6^ cells/mL and transferred to a new medium after reaching the stationary growth phase, twice a week.

#### 2.7.2. Infection and Treatment Protocols

BALB/c mice were first infected with 5 × 10^6^ stationary phase promastigotes of *L. amazonensis* intradermally at the tail base. As illustrated in Appendix A, the lesion appeared progressively as a function of time, starting with a closed lesion that became ulcerated in most animals after 70 days. Three experiments were performed, with treatment initiated 75 days post-infection.

In the first experiment, the formulations of AmB in conventional and PEGylated liposomes were compared for their efficacy by intraperitoneal (IP) route, following seven doses of 5 mg/kg given at 4-day intervals. Animals were divided into the five following groups (n = 8–10/group): CONV-LAmB group, receiving the conventional liposomal AmB formulation; PEG-LAmB group, receiving the PEGylated liposomal AmB formulation; Miltefosine group, receiving miltefosine by oral route at 10 mg/kg daily for 24 days; Empty-Lipo group, receiving by IP route a mixture of empty conventional and PEGylated liposomes at 1:1 lipid mass ratio and the same regimen as in CONV-LAmB group; Saline group, receiving isotonic saline by IP route.

In the second experiment, the formulations of AmB in conventional and PEGylated liposomes were compared to AmBisome^®^ for their efficacy by intravenous (IV) route, following seven doses of 5 mg/kg given at 4-day intervals. Animals were divided into the five following groups (n = 7–10/group): CONV-LAmB group, receiving the conventional liposomal AmB formulation; PEG-LAmB group, receiving the PEGylated liposomal AmB formulation; AmBisome^®^ group, receiving AmBisome^®^; Empty-LCONV group, receiving by IV route empty conventional liposomal formulation at the same regimen (lipid dose, intervals) as that in CONV-LAmB group; Empty-LPEG group, receiving by IV route empty PEGylated liposomal formulation at the same regimen (lipid dose, time intervals) as in PEG-LAmB group.

In the third experiment, the PEGylated liposomal AmB formulation was evaluated for its efficacy by oral route following ten doses of 5 mg/kg given at 2-day intervals and compared to the same formulation and AmBisome^®^ given by IP route at the same dosage and time intervals. Animals were divided into the five following groups (n = 10/group): PEG-LAmB (oral) group, receiving the PEGylated liposomal AmB formulation by the oral route; PEG-LAmB (IP) group, receiving the PEGylated AmB liposomal AmB formulation by IP route; AmBisome^®^ (IP) group, receiving AmBisome^®^ by IP route; Miltefosine group, receiving miltefosine at 10 mg/kg, every 2 days by the oral route; Saline group, receiving isotonic saline by IP route.

#### 2.7.3. Evaluation of Treatment Efficacy

The size of the lesion was determined, either at 4-day intervals or weekly, using an analog universal caliper, 150 mm, Digimess^®^ (Brazil), with an accuracy of 0.05 mm. The estimated lesion size was obtained by measuring horizontal (*L*_H_) and vertical (*L*_V_) lengths of the lesion, respectively, perpendicular and parallel to the vertebral column of the animal, and the average lesion size (= (*L*_H_ + *L*_V_)/2) was calculated each time for each animal. Since *L*_H_ and *L*_V_ did not differ significantly in the infected animals, the lesions could be considered approximately circular. The lesion size growth was calculated in each animal as the difference in the average lesion size between time t and day zero of treatment.

Three days after the end of treatment, mice were anesthetized with xylazine 8 mg/kg and ketamine 75 mg/kg, then euthanized by cervical dislocation and the lesion and the spleen were collected for evaluation of the parasitic load by qPCR, as described previously [20]. Briefly, the tissue samples were macerated, and an aliquot was added to a microtube containing lysis buffer and Proteinase K, vortexed and incubated at 56 °C overnight. After incubation, DNA was extracted with NucleoSpin^®^ Tissue Kit (MN, Macherey-Nagel GmbH & Co. KG, Dürin, Germany), according to the manufacturer’s instructions. The DNA concentrations were measured by spectrophotometry (Abs at 280/260 nm) and adjusted to 20 ng/μL. One microliter of each sample was used to a final volume of 20 μL per reaction that included ultrapure water, 5.0 μL of SYBR^®^ Green PCR Master Mix (Warrington, UK), 10 pmol of each oligonucleotide, as sense (forward, 5′-CGT GGG GGA GGG GCG TTC T-3′ R: 5′-CCG AAG CAG CCG CCC CTA TT-3′) and antisense primers (reverse, 5′-CCG AAG CAG CCG CCC CTA TT-3′) constructed for amplification of the mini-circle region present in the kinetoplast DNA (kDNA) of approximately 120 bp. The standard curve was constructed with serial dilutions of the known suspension of *Leishmania amazonensis* (IFLA/BR/1967/PH8), in the range of 10^1^ to 10^8^ parasites, submitted to extraction. The amplification protocol included an annealing temperature and extension of 60 °C, with melting curve construction, on the Applied Biosystems™ 7500 Fast Real-Time PCR System (Thermo Fisher Scientific, Waltham, MA, USA) and the analysis was made using the 7500 System Software. Data are presented as the number of *Leishmania* per ng of total DNA.

#### 2.7.4. Evaluation of Toxicity

The animal groups receiving the formulations of AmB in the third experiment were evaluated for changes in the levels of plasma markers of renal and hepatic functions, in comparison to the control group. Blood (0.5–1 mL) was collected by the orbital plexus in a BD Vacutainer^®^ tube containing EDTA as an anticoagulant, just after animal anesthesia and before euthanasia. Plasma was separated by centrifugation and stored at −20 °C. Hepatic injury was evaluated through the determination of the enzymatic activity of alanine aminotransferase (ALT) and alkaline phosphatase (ALP). Renal toxicity was evaluated through the determination of urea and creatinine levels. These determinations were performed using commercial kits (Bioclin Quibasa, Belo Horizonte, Brazil) and following the manufacturer’s instructions.

### 2.8. Statistical Analyses

The one-way ANOVA with Tukey’s post-test (for normally distributed data) or the Kruskal–Wallis non-parametric test with Dunn’s post-test were used for statistical analyses of parasite load and serum levels of urea and creatinine, with significance level *p* < 0.05. The normal distribution was checked with the following tests: the Anderson–Darling test, the D’Agostino and Pearson test and the Shapiro–Wilk test. Homoscedasticity was checked using the Brown–Forsythe test. Two-way ANOVA (repeated measures) was used to compare the variation in lesion size between the experimental groups, followed by Dunnett’s post-test. *p* < 0.05, *p* < 0.01, *p* < 0.001 and *p* < 0.0001 were marked with *, **, *** and ****, respectively. The graphics and statistical analyses were performed using GraphPad Prism^®^ (version 9) software (GraphPad Software, San Diego, CA, USA).

## 3. Results

### 3.1. Physicochemical Characterization of Liposomal Formulations of AmB

In this work, a simple and original process has been developed for the encapsulation of AmB in pre-formed empty liposomes that was successfully applied to conventional liposomes with the same lipid composition as AmBisome^®^ and, also, to PEGylated liposomes.

Table 1 displays the results of particle size distribution, zeta-potential and drug encapsulation efficiency characterizations. The formulations of AmB in conventional and PEGylated liposomes (CONV-LAmB and PEG-LAmB) showed nanoparticles with mean hydrodynamic diameters lower than 130 nm and narrow size distribution (PI < 0.3). PEGylation of liposomes resulted in a sharp reduction in the zeta-potential, as expected from the coating of nanoparticle surface by hydrophilic PEG polymer. AmB was incorporated with an EE greater than 97%, an AmB total content close to 90%, and little influence on the particle size distribution and zeta-potential. The particle diameter of the new formulations was significantly greater than that of AmBisome^®^. Cryo-TEM analyses (Figure 1) further confirmed the vesicular morphology of nanoparticles and the lower size of AmBisome^®^.

The state of aggregation of AmB in the conventional and PEGylated liposomes was assessed by absorption and CD spectroscopies. As shown in Figure 2, the new formulations exhibited very similar absorption profiles with four bands: an intense band centered at 328 nm and three bands of decreasing intensities at 363 nm, 388 nm and 415 nm. The absorption spectrum of AmBisome^®^ was close to those of the new formulations, except for a blue shift of the most intense band (centered at 323 nm). The CD spectra of the new formulations were also very similar, showing a positive band at 328 nm, followed by a negative band at 345 nm. This is in contrast with the CD spectrum of AmBisome^®^ that showed an intense couplet-type dichroic signal centered at 329 nm (with a positive Cotton effect at 323 nm and a negative Cotton effect at 334 nm), which is characteristic of an aggregated form of AmB. The spectra of the liposomal formulations also markedly differed from those of AmB prepared in PBS (without liposomes using the same process of preparation of the liposomal formulations) showing a lower UV absorption and a couplet-type dichroic signal.

The absence of a couplet-type signal in the CD spectra of the new formulations reinforces the model of incorporation of AmB in the membrane of liposomes under the non-aggregated form, as AmB in the aqueous phase self-associates and exhibits a couplet-type signal. The fact that the absorption spectra of the formulations also differed from that of AmB monomeric form in ethanol [21], which shows a predominant band around 410 nm, also brings strong evidence that the incorporation in liposomes is mediated by the specific interaction of AmB with membrane lipid(s). Importantly, this study establishes that the aggregation of AmB in the new formulations differs from that in AmBisome^®^.

The process used here to incorporate AmB into pre-formed liposomes involves two steps: a first step consisting of the addition of AmB dissolved in NaOH 0.1 M to the liposome suspension, followed by adjustment of pH to 6.5 after 2 min of incubation; the second step consisting in 5-min heating of the drug–liposome mixture at a temperature greater than the phospholipid phase transition temperature. The importance of the second step to promote the complete incorporation of AmB in the membrane was evidenced by the changes induced in the CD spectra (Figure 3). It is also noteworthy that the spectra obtained after the heating step were similar to those registered after freeze-drying and reconstitution with water, indicating that freeze-drying preserved the final state of AmB in the membrane.

As this work aims to investigate the efficacy of the novel liposomal AmB formulations by parenteral and oral routes, we also evaluated the changes in the absorption and CD spectra of AmB following dilution of the liposomal formulations in either PBS or a simulated gastric fluid and incubation for 2 h at 37 °C. As shown in Figure 4, exposition to the acidic medium promoted marked changes in the absorption and CD spectra of AmB in the conventional formulation, but not in the PEGylated formulation. The spectral differences between the neutral and acidic media for the conventional formulation were also accompanied by a change in the particle size distribution (diameter = 149 nm and PI = 0.48 in HCl vs. diameter = 141 nm and PI = 0.33 in PBS). On the other hand, no difference in particle size was observed for the PEGylated formulation (diameter = 159 nm and PI = 0.18 in HCl vs. diameter = 159 nm and PI = 0.19 in PBS). The apparently greater stability of the PEGylated liposomal AmB in the acid medium led us to choose this formulation for evaluation by oral route.

### 3.2. Antileishmanial Efficacy of Liposomal AmB Formulations in Murine Model of CL

Figure 5 shows the efficacy of treatment of *L. amazonenzis*-infected BALB/c mice with conventional and PEGylated liposomal AmB formulations given every four days, either by IP (a, b) or IV route (c, d). The evaluated parameters were the growth of lesion size and the parasite load in the lesion after 24 days of treatment. In this model, treatment with oral miltefosine as a positive control, given at 10 mg/kg daily promoted marked decreases in the lesion size and parasite load. In both studies, PEGylated liposomal AmB formulation either by IP or IV route promoted a significant reduction in lesion size growth, when compared to the control (either saline or empty liposomes). The clinical efficacy of this formulation was confirmed by the significant suppression of parasites in the lesion, in comparison with controls. This is in contrast with the lack of significant efficacy of the conventional liposomal AmB formulation (given either IP or IV) and AmBisome^®^ (given IV) in this experimental model and specific treatment regimen. The same profile of parasite suppression was observed in the spleen of animals, with a significant reduction in parasite load only in the group that received PEGylated liposomal AmB formulation by IV route (Appendix A).

In a subsequent experiment, the antileishmanial efficacy of PEGylated liposomal AmB was evaluated by the oral route and compared to those of the same formulation and AmBisome^®^ given by the IP route. In this study, infected mice were treated with 10 doses of 5 mg/kg given at 2-day intervals. As shown in Figure 6, the PEGylated formulation promoted significant reductions in lesion size growth and parasite load, to comparable levels to those achieved with AmBisome^®^ (IP). An evaluation of the parasite load in the spleen showed significant parasite suppression only in the groups that received the formulations by IP route (Appendix A). In this experiment, the groups that received liposomal AmB formulations were further evaluated, regarding plasma markers of renal (urea, creatinine) and hepatic (ALT and ALP) functions and injuries, in comparison to the control group. Only urea showed significant change, with significantly increased levels in AmBisome^®^ (IP) and PEG-LAmB (IP) groups (Appendix A).

## 4. Discussion

Among the few drugs available for leishmaniasis, AmB is considered the most potent antileishmanial agent currently available and the least susceptible to the emergence of resistance [11]. However, its low solubility and tendency to self-aggregate result in high toxicity and low oral drug bioavailability. This context has stimulated the search for formulations in which AmB is incorporated under the less toxic non-aggregated form [10,11]. The development of such formulations is a great challenge due to the difficulty in controlling the interactions of AmB with its carriers, considering the complex structure of AmB and its amphoteric and amphiphilic characters [22]. The present work presents an innovative process for incorporating AmB into pre-formed liposomes, which exploits the effect of pH on the solubility of AmB and the influence of temperature on membrane fluidity and insertion of AmB into the liposomal membrane. The AmB molecule exhibits carboxyl and an amino group, with pKa of 5.7 and 10.0, respectively. In a previous study, the formation of AmB aggregates in water was evidenced at acidic and neutral pH values, confirming that either the protonated form of the carboxylic group or the positive net charge at the amino group participates in the stabilization and formation of aggregates [21]. On the other hand, when raising the pH to values >10, the deprotonation of the amine group displaced the equilibrium to the monomeric form. Thus, the first step of incubation of AmB with empty liposomes at basic pH is critical as it most probably promotes the interaction of the monomeric form with the lipids. As a significant advantage of our preparation process in comparison to conventional methods, liposome size calibration is performed in the absence of AmB and no organic solvent is used for incorporating AmB into liposomes. Thus, novel formulations of AmB were achieved with conventional and PEGylated liposomes, showing high drug encapsulation efficiencies and adequate particle size distributions for parenteral administration. The formulation of AmB in conventional liposomes had the same lipid composition as the commercial liposomal AmB formulation (AmBisome^®^). However, a major physicochemical difference refers to the incorporation of AmB under non-aggregated form in both the conventional and PEGylated liposomes. This difference most probably arises from the specific process used here for the incorporation of AmB, which seems to favor the formation of a drug–lipid complex with a single AmB molecule.

As another important contribution, this study reveals the distinct therapeutic profile of the PEGylated liposomal formulation, in comparison to AmBisome^®^.

First, the PEGylated formulation exhibited greater therapeutic efficacy than AmBisome^®^, when given by parenteral route in murine CL. A previous study has evidenced that AmBisome^®^ delivered AmB more effectively to the skin lesion in the murine CL model than liposomes of larger size [9]. It is noteworthy that our PEGylated liposomes exhibit larger sizes than AmBisome^®^. Thus, the superiority of our formulation most probably arises from the PEG-mediated long-circulating characteristics of these vesicles and their enhanced extravasation through the leaky capillaries in the inflamed lesion skin. Although it is well established that PEGylation of neutral liposomes with 5 mol% of DSPE-PEG2000 enhances their blood circulation time [23], that is not so clear for negatively charged liposomes. A previous study showed that liposomes containing either phosphatidylserine or phosphatidylglycerol, both as conventional or PEGylated formulations, were rapidly cleared from the circulation [24]. On the other hand, a different profile was observed by our group using anionic liposomes containing dicetylphosphate, with or without 5 mol% of DSPE-PEG2000. The PEGylated liposomes promoted a prolonged circulation time of encapsulated antimonial drug, in comparison with non-PEGylated liposomes, and were more effective in reducing the skin parasite load in canine leishmaniasis [25,26]. Nevertheless, it is possible that distinct drug release profiles from liposomes may contribute to the therapeutic difference between the PEGylated formulation and AmBisome^®^. The precise mechanism(s) responsible for the greater efficacy of PEGylated liposomal AmB should be elucidated in future studies through evaluations of the drug release kinetics and pharmacokinetics. As the present study was performed exclusively with female mice, another aspect that deserves investigation is the therapeutic efficacy of our formulation in *Leishmania*-infected males. Indeed, sex-related differences in the manifestation of infections with Leishmania species and in rates of treatment failure or adverse effects have been reported [27]. To the best of our knowledge, this study is the first to report a liposomal formulation of AmB showing greater efficacy than AmBisome^®^ in a model of CL. Indeed, previous work has described a novel liposomal formulation of AmB that was as effective as AmBisome^®^ in the murine model of Old World CL [28]. Our data further suggest the great potential of the PEGylated formulation for the treatment of disseminated infections, comprising not only leishmaniases but also life-threatening systemic fungal infections.

Surprisingly, the PEGylated liposomal AmB formulation also exhibited therapeutic efficacy by the oral route in murine CL. The oral efficacy was achieved using a relatively low dose of AmB (5 mg/kg), given on alternate days. This suggests that treatment efficacy may be further enhanced with a daily regimen or by increasing the dose. This is a pioneer study, as it reports for the first time an orally effective formulation of AmB for the treatment of CL. Previous studies have reported the development of lipid formulations of AmB for the treatment of VL and fungal infections [10], but none has described an effective liposomal formulation. Our work also suggests that PEGylation improved the stability of the liposomal AmB formulation in an acidic environment and protected the drug molecule from acid degradation. In agreement with this model, a previous study reported that self-assembling lecithin-based mixed polymeric micelles incorporating DSPE-PEG showed enhanced oral bioavailability of AmB [29]. An additional advantage of an oral formulation of AmB is its expected reduced toxicity with respect to parenteral formulations [10], highlighting its potential to improve patient compliance and quality of life. Indeed, the reduced renal toxicity of our oral liposomal formulation is supported here by the absence of change in the plasma level of urea, in contrast to the parenteral formulations (AmBisome^®^ and PEGylated liposomal AmB).

## 5. Conclusions

The present study reports a unique PEGylated liposomal formulation of AmB that is more effective than AmBisome^®^ against murine CL by both parenteral and oral routes. Considering that CL is a neglected tropical disease, the new formulation could bring great benefits in improving the access of patients to more effective and safer treatments.

## 6. Patents

The following patent application was deposited in Brazil on 9 April 2021: Frézard, F.; Ramos, G.S.; Ferreira, L.A.M.; Fujiwara, R.T.; Vallejos, V.M.R.; Borges, G.S.M. Processo para obtenção de lipossomas conjugados a anfotericina B, formulação e usos. BR1020210068205.

## Figures and Tables

**Figure 1 pharmaceutics-14-00989-f001:**
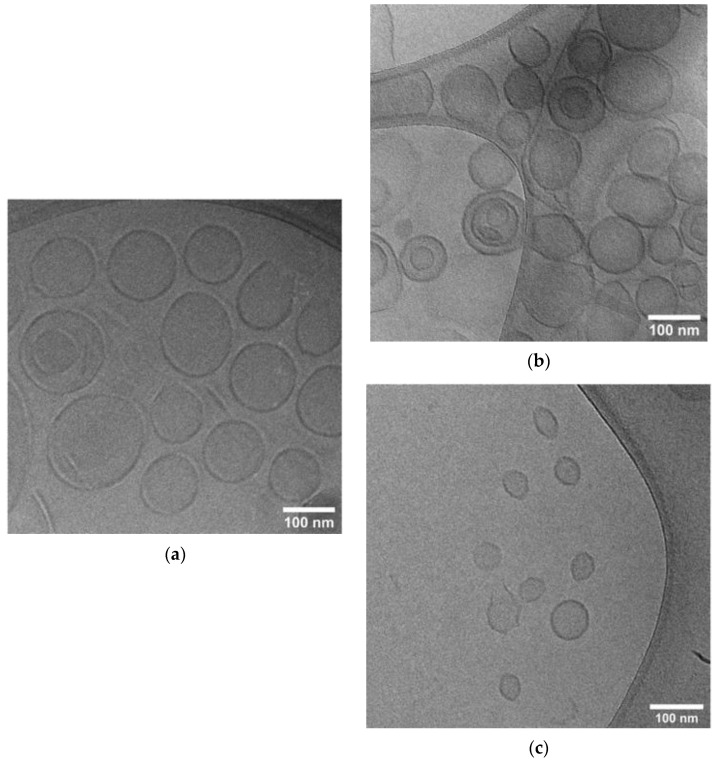
Cryo-TEM images of different liposomal formulations of AmB: (**a**) PEGylated liposomal formulation; (**b**) conventional liposomal formulation; (**c**) AmBisome^®^.

**Figure 2 pharmaceutics-14-00989-f002:**
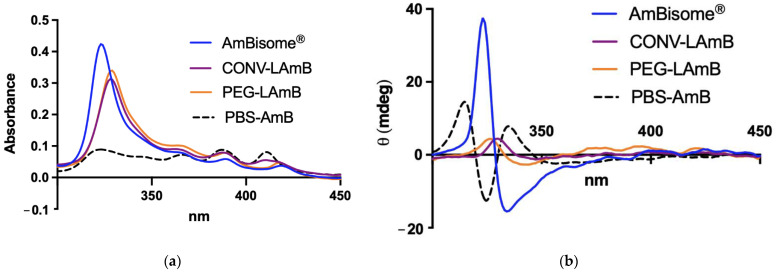
Characterization of AmB aggregation state by: (**a**) spectrophotometry and (**b**) circular dichroism in conventional (CONV-LAmB) and PEGylated (PEG-LAmB) liposomal AmB, in comparison to AmBisome^®^. PBS-AmB is a solution of AmB prepared in PBS without liposomes, using the same process of preparation of the liposomal formulation, except for the freeze-drying step.

**Figure 3 pharmaceutics-14-00989-f003:**
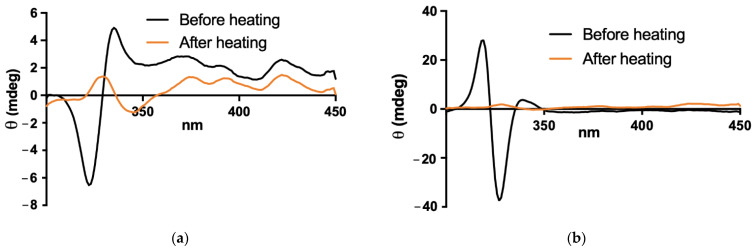
Circular dichroism spectra of AmB in the presence of pre-formed (**a**) PEGylated or (**b**) conventional liposomes, before and after heating at 60 °C, as second step of the process of incorporation of AmB.

**Figure 4 pharmaceutics-14-00989-f004:**
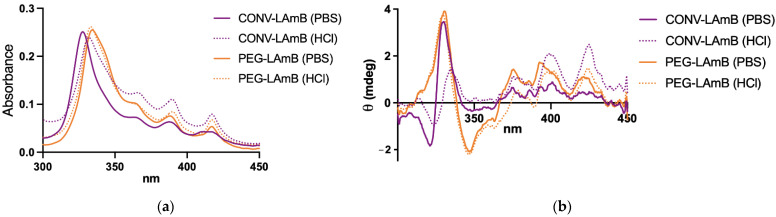
Absorption (**a**) and circular dichroism (**b**) spectra of AmB in PEGylated and conventional liposomal formulations, after 10-fold dilution in a medium simulating the gastric fluid or PBS and incubation for 2 h at 37 °C. Spectra were registered at 25 °C after 100-fold dilution in PBS.

**Figure 5 pharmaceutics-14-00989-f005:**
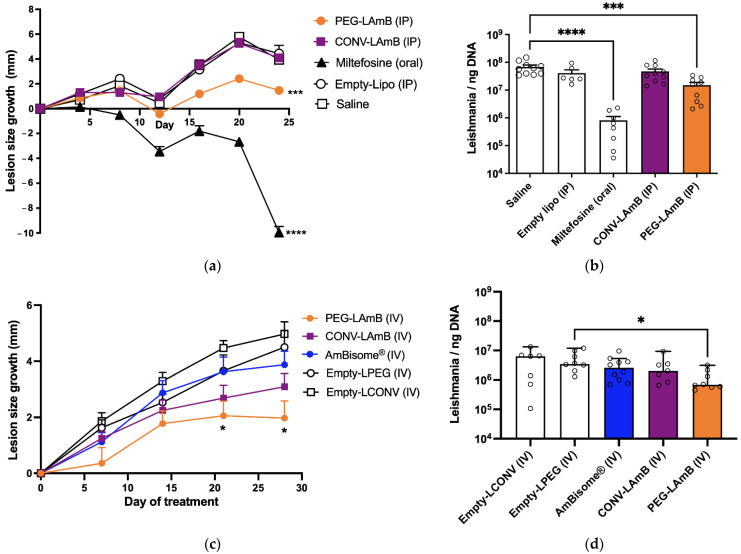
Antileishmanial efficacies of conventional and PEGylated liposomal AmB by IP (**a**,**b**) or IV (**c**,**d**) routes in murine model of cutaneous leishmaniasis. BALB/c mice were infected intradermally with *Leishmania amazonensis*. Treatments were initiated 75 days after infection with 7 doses of the following formulations. (**a**,**b**) CONV-LAmB group received the conventional liposomal AmB formulation at 5 mg/kg, every 4 days by IP route; PEG-LAmB group received the PEGylated liposomal AmB formulation at 5 mg/kg, every 4 days by IP route; Miltefosine group received miltefosine by oral route at 10 mg/kg daily for 24 days; Empty-Lipo group received by IP route a mixture of empty conventional and PEGylated liposomes at 1:1 lipid mass ratio and the same regimen as in CONV-LAmB group; Saline group received isotonic saline by IP route. (**c**,**d**) CONV-LAmB group received the conventional liposomal AmB formulation at 5 mg/kg, every 4 days by IV route; PEG-LAmB group received the PEGylated liposomal AmB formulation at 5 mg/kg, every 4 days by IV route; AmBisome^®^ group received AmBisome^®^ at 5 mg/kg, every 4 days by IV route; Empty-LCONV group received by IV route empty conventional liposomal formulation at the same regimen (lipid dose, intervals) as that in CONV-LAmB group; Empty-LPEG group received by IV route empty PEGylated liposomal formulation at the same regimen (lipid dose, time intervals) as in PEG-LAmB group. Treatment efficacy was evaluated through measurement of lesion size growth (**a**,**c**), analyzed by two-way ANOVA for repeated measures) and determination of parasite load by qPCR (**b**) and (**d**). (**b**) Data shown as means + SEM and analyzed through one-way ANOVA followed by Dunnett’s multiple comparison post-test. (**d**) Data shown as medians + 95% confidence interval and analyzed by Kruskal-Wallis followed by Dunn’s multiple comparison post-test. ** p* < 0.05, **** p* < 0.001 and **** *p* < 0.0001.

**Figure 6 pharmaceutics-14-00989-f006:**
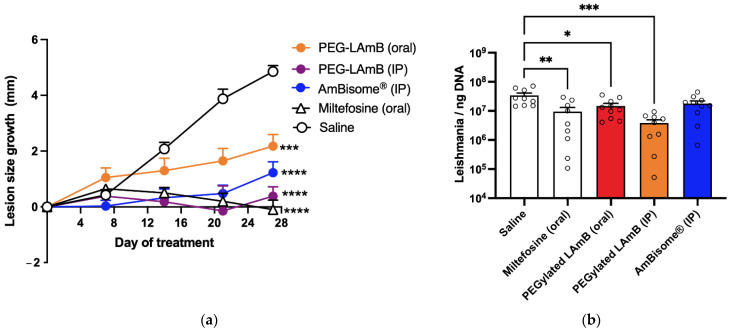
Antileishmanial efficacies of PEGylated liposomal AmB by oral route in comparison with IP route in murine model of cutaneous leishmaniasis. BALB/c mice were infected intradermally with *Leishmania amazonensis*. Treatments were initiated 75 days after infection with 10 doses of the following formulations. PEG-LAmB (oral) group received the PEGylated liposomal AmB formulation at 5 mg/kg, every 2 days by oral route; PEG-LAmB (IP) group received the PEGylated AmB liposomal AmB formulation at 5 mg/kg, on alternate days by IP route; AmBisome^®^ (IP) group received AmBisome^®^ at 5 mg/kg, every 2 days by IP route; Miltefosine group received miltefosine at 10 mg/kg, on alternate days by oral route; Saline group received isotonic saline by IP route. Treatment efficacy was evaluated through measurement of lesion size growth (**a**), analyzed by two-way ANOVA for repeated measures) and determination of parasite load by qPCR (**b**). (**b**) Data are shown as means + SEM and analyzed through one-way ANOVA followed by Tukey’s multiple comparison post-test. ** p* < 0.05, ** *p* < 0.01, **** p* < 0.001 and **** *p* < 0.0001.

**Table 1 pharmaceutics-14-00989-t001:** Particle size distribution, zeta-potential, drug encapsulation efficiency (EE), and AmB total content of PEGylated (PEG-LAmB) and conventional (CONV-LAmB) liposomal AmB, in comparison to AmBisome^®^.

Formulation	Diameter (nm) ± SD ^1,2^	Polydispersity Index ± SD ^1,2^	Zeta Potential (mV) ± SD ^1,3^	EE ± SD ^1,3^	AmB Total Content (mg/mL) ± SD ^1,3^ (%AmB Recovery)	Final Loading Content AmB/Lipid (*w*/*w*)
PEG-LAmB	125 ± 12	0.23 ± 0.03	−3.4 ± 0.3	97.9 ± 0.7%	3.76 ± 0.06 (94%)	0.118
CONV-LAmB	113 ± 15	0.09 ± 0.04	−27.1 ± 0.1	104.9 ± 1.3%	3.59 ± 0.04 (90%)	0.133
AmBisome^®^	89 ± 6	0.15 ± 0.02	−30 ± 1	-	-	-
Empty-LPEG	134 ± 10	0.18 ± 0.02	−4.7 ± 0.4	-	-	-
Empty-LCONV	103 ± 2	0.05 ± 0.03	−28.1 ± 0.6	-	-	-

^1^ SD: standard deviation; ^2^ Mean and SD from 5 independent batches (except for AmBisome^®)^ with measurements of 8 samples from one batch); ^3^ Mean and SD from 3 independent batches.

## Data Availability

Data is contained within the Appendix A.

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
