# Peer review of "Formulation of Amphotericin B in PEGylated Liposomes for Improved Treatment of Cutaneous Leishmaniasis by Parenteral and Oral Routes"

_pharmaceutics, 2022, doi:10.3390/pharmaceutics14050989_

Round 1
Reviewer 1 Report
With interest I read your manuscript about liposomes & AmB. Before I can recommend it for publication, the following issues have to be addressed.
1) 2.7.1. Animals: Why did you use only female mice?
2) 2.7.2. Why do you have different group sizes (n=7-10 or 8-10) in your in vivo study? And did you take this into account when analysing and evaluating your data?
3) 2.7.3. You stated that you measured the lesion size with a caliper. Could you add the precision of the instrument? Then, you wrote that the average size was calculated. Equation? The background behind that is that the determined values as well as the on that basis calculated values have errors and that has to be taken into account when comparing group results.
4) 2.7..4 How did you collected your blood samples? Volume? How was the plasma seperated? Any anti-coagulation?
5) 2.8. How did you checked the normal distribution?
6) Table 1. Why do you have varying N (or n?)? Regarding EE, why do you gained >100% in CONV-LAmB or in other words what is the 100% reference to that?
7) In lines 345 & 347 you present size values in PBS divergent from the values in Table 1. Why or did I miss understood you here?
8) 3.2. You compared sometimes the means and sometimes the medians of your group values. Why did you choose which value in the corresponding data set? Also, the variation of the data is shown by different errors or even confidential intervals.
9) Figure 5. How was the group composition on d0 of treatment? This is realy important to judge the following monitoring data. Did the lesion size increased continously before starting treatment? I would prefer that you present also the data before treatment to get a better impression of the lesion growth curve without and later with treatment. Treatment should be visible in the graph, e.g. indicated by arrows. What does these lots of stars (*/***/****) mean (this is also relevant for other figures)? Under b) you wrote means +/- SEM in the plot it's just + SEM?! Why is the scaling between b) and d) different? Because of that a comperison by one glance is a bit hindered.
10) Why do you think/state that CL is a neglected tropical desease?
Reviewer 2 Report
The paper “Formulation of amphotericin B in PEGylated liposomes for improved treatment of cutaneous leishmaniasis by parenteral and oral routes” focuses on the treatment of cutaneous leishmaniasis by using pegylated liposomes. The paper is interesting and well organized, however some minor points should be addressed before to be accepted for publication in this journal.
What has caught my attention is the presence of miltefosine in the materials? Is it a mistake or it has a role in this study?
The choice of the preparation method should be better discussed along with the need of basic environment.
Please carefully check the English language throughout the manuscript
Author Response
Please, find below our responses to your comments.
The paper “Formulation of amphotericin B in PEGylated liposomes for improved treatment of cutaneous leishmaniasis by parenteral and oral routes” focuses on the treatment of cutaneous leishmaniasis by using pegylated liposomes. The paper is interesting and well organized, however some minor points should be addressed before to be accepted for publication in this journal.
What has caught my attention is the presence of miltefosine in the materials? Is it a mistake or it has a role in this study?
As mentioned in the introduction (line 55), miltefosine is one of the few drugs available for treatment of cutaneous leishmaniasis. In fact, it also used in Brazil for the treatment of canine visceral leishmaniasis. Therefore, we used oral miltefosine as positive control in the first in vivo experiment (Figure 5 a and b).
The choice of the preparation method should be better discussed along with the need of basic environment.
To address the Reviewer concern, the following paragraph was added to the Discussion (lines 499-507) to give mechanistic insight into the role of basic environment:
“The AmB molecule exhibits carboxyl and an amino group, with pKa of 5.7 and 10.0, respectively. In a previous study, the formation of AmB aggregates in water was evidenced at acidic and neutral pH values, confirming that either the protonated form of the carboxylic group or the positive net charge at the amino group participates in the stabilization and formation of aggregates [21]. On the other hand, when raising the pH to values > 10, the deprotonation of the amine group displaced the equilibrium to the monomeric form. Thus, the first step of incubation of AmB with empty liposomes at basic pH is critical as it most probably promotes the interaction of the monomeric form with the lipids.”
Please carefully check the English language throughout the manuscript
The English language will be revised, by applying to the service of MDPI.
The authors thank the Reviewer for his valuable recommendations and comments that help improving our manuscript.
Reviewer 3 Report
In this work authors described the amphotericin B in PEGylated liposomes for improved treatment of cutaneous leishmaniasis by parenteral and oral routes. The work is well presented and I suggest the publication after minor reviosions
- Table 1, loading content of AmB should be calculated and added.
- Why 5% of DSPE-PEG2000 was chosen for this study?
- Line 446-448, several studies (doi.org/10.3390/pharmaceutics13020282;doi.org/10.1016/j.actbio.2020.11.049) related to this long-ciruclation point should be included.
- A section of conclusion should be seperated from the discussion.
Author Response
Please, find below our responses to your comments.
In this work authors described the amphotericin B in PEGylated liposomes for improved treatment of cutaneous leishmaniasis by parenteral and oral routes. The work is well presented and I suggest the publication after minor revisions
- Table 1, loading content of AmB should be calculated and added.
Loading content of AmB was calculated as (encapsulated AmB/lipid) mass ratio. The results were added to Table 1. Information was also added in the method section (2.4).
- Why 5% of DSPE-PEG2000 was chosen for this study?
Line 446-448, several studies (doi.org/10.3390/pharmaceutics13020282;doi.org/10.1016/j.actbio.2020.11.049) related to this long-circulation point should be included.
5 mol% of DSPE-PEG2000 is classically used in liposomal formulations to achieve long-circulating properties. This corresponds to the mol% of DSPE-PEG2000 in the commercial long-circulating liposomal formulation Doxil®. Our group have also previously studied anionic PEGylated liposomes (containing 5 mol% of DSPE-PEG2000). The PEGylated liposomes promoted prolonged circulation time of encapsulated antimonial drug, in comparison with non-PEGylated liposomes, and were more effective in reducing the skin parasite load in canine leishmaniasis [Azevedo et al., 2014; Dos Santos et al., 2021].
The reference doi.org/10.3390/pharmaceutics13020282 was added to the Discussion, as proposed by the Reviewer. On the hand, although very interesting, reference doi.org/10.1016/j.actbio.2020.11.049 could not incorporated, as it was too far from our main subject.
To address the Reviewer concern, the following paragraph was added to Discussion:
“Although it is well established that PEGylation of neutral liposomes with 5 mol% of DSPE-PEG2000 enhances their blood circulation time [23], this is not so clear for negatively charged liposomes. A previous study showed that liposomes containing either phosphatidylserine or phosphatidylglycerol, both as conventional or PEGylated formulations, were rapidly cleared from the circulation [24]. On the other hand, a different profile was observed by our group using anionic liposomes containing dicetylphosphate, with or without 5 mol% of DSPE-PEG2000. The PEGylated liposomes promoted prolonged circulation time of encapsulated antimonial drug, in comparison with non-PEGylated liposomes, and were more effective in reducing the skin parasite load in canine leishmaniasis [25,26].”
A section of conclusion should be separated from the discussion
The new section was created in the revised version of our manuscript.
The authors thank the Reviewer for his valuable recommendations and comments that help improving our manuscript.
Round 2
Reviewer 1 Report
Thank you for your revision and answers.
Just three little things left:
1) Fig. 5b was modified for better comparison. That makes it easier. But I would also set both y axes in the same range not only same type of scaling. Since this two results should be compared directly with each other, right?
2) Why did you choose this equation for lesion size measurement? I just wonder because this is not an area and I would think it should be, or? If you determine two lengths, you can calculate an lesion area (ellipse based e.g.?), or?
3) In your answer to my 2nd question you stated, that different groups sizes do not matter or the tests take this into account, resp. Therefore, I assume that you checked also for heteroscedasticity for the ANOVA, right? Because this could really influence the analysis, esp. with varying group sizes...
And two small final comments:
I am not convinced that ***/**** (and look-alikes) really improve data/results. It looks sometimes a bit like "star collecting". For sure, significance is useful but do not overtax it, esp. when dealing with small & varying sample sizes.
Easier handling & experimental setups do not justify the ignorance of approx. half of a population in my opinion. And pharmacokinetics as well as -dynamics (can) vary inside the sexes. Not to speak of clinical differences in course of disease, which might be also influenced by sex. In addition, the littermates unavoidably include both sexes - what did you do with the non-used ones? Hopefully, they are used wisely also.
Author Response
1) Fig. 5b was modified for better comparison. That makes it easier. But I would also set both y axes in the same range not only same type of scaling. Since this two results should be compared directly with each other, right?
Response: Thank you for the recommendation. Fig. 5d has been adjusted with the same range as Fig 5b. To further maintain homogeneity the scale of Fig. 6b was also adjusted with same range.
2) Why did you choose this equation for lesion size measurement? I just wonder because this is not an area and I would think it should be, or? If you determine two lengths, you can calculate an lesion area (ellipse based e.g.?), or?
Response: We agree with Reviewer that this is an important point. We chose the equation for lesion size measurement, since the horizontal and vertical lengths did not differ significantly (according to paired t test). Thus, the lesions could be considered as approximately circular. In this case, representing the lesion size as the mean diameter seems acceptable. This point has been clarified in the method section (2.7.3.).
3) In your answer to my 2nd question you stated, that different groups sizes do not matter or the tests take this into account, resp. Therefore, I assume that you checked also for heteroscedasticity for the ANOVA, right? Because this could really influence the analysis, esp. with varying group sizes...
Response: We agree with Reviewer that this is another important point. Homoscedasticity was checked using the Brown-Forsythe test. This information has been added in the Method section (2.8.).
4) I am not convinced that ***/**** (and look-alikes) really improve data/results. It looks sometimes a bit like "star collecting". For sure, significance is useful but do not overtax it, esp. when dealing with small & varying sample sizes.
Response: To make the explanation “lighter”, the information was moved to the Method section (2.8.) and removed from individual figure legends.
5) Easier handling & experimental setups do not justify the ignorance of approx. half of a population in my opinion. And pharmacokinetics as well as -dynamics (can) vary inside the sexes. Not to speak of clinical differences in course of disease, which might be also influenced by sex. In addition, the littermates unavoidably include both sexes - what did you do with the non-used ones? Hopefully, they are used wisely also.
Response: We agree with the Reviewer that this point is relevant, as there is scientific evidence of sex-related difference in manifestation to infections with leishmania species and in rates of treatment failure or adverse effects (see reference below).
Lockard RD, Wilson ME, Rodríguez NE. Sex-Related Differences in Immune Response and Symptomatic Manifestations to Infection with Leishmania Species. J Immunol Res. 2019 Jan 10;2019:4103819. doi: 10.1155/2019/4103819
A paragraph with the citation of this reference has been added to the Discussion. in response to Reviewer’s concern.
The english language has been revised throughout the entire manuscript.
The authors thank the Reviewer for his valuable comments that help improving greatly our manuscript.